# Characterisation of Low Molecular Weight Compounds of Strawberry Tree (*Arbutus unedo* L.) Fruit Spirit Aged with Oak Wood

Ofélia Anjos [1,2,3,*] , Carlos A. L. Antunes [4] , Sheila Oliveira-Alves [2,5] , Sara Canas [5,6] and Ilda Caldeira [5,6]

1 CERNAS-IPCB, Research Centre for Natural Resources, Environment and Society, Polytechnic Institute of Castelo Branco, 6001-909 Castelo Branco, Portugal
2 CEF—Forest Research Centre, School of Agriculture, University of Lisbon, 1349-017 Lisboa, Portugal; sheila.alves@iniav.pt
3 Spectroscopy and Chromatography Laboratory, Centre of Plant Biotechnology of Beira Interior, 6001-909 Castelo Branco, Portugal
4 APROSER—Associação de Produtores do Concelho da Sertã, 6100-601 Sertã, Portugal; carlosalbertoantunescb@gmail.com
5 INIAV—Polo de Inovação de Dois Portos/Estação Vitivinícola Nacional, Quinta da Almoínha, 2565-191 Dois Portos, Portugal; sara.canas@iniav.pt (S.C.); ilda.caldeira@iniav.pt (I.C.)
6 MED—Mediterranean Institute for Agriculture, Environment and Development, University of Évora, Pólo da Mitra, Ap. 94, 7006-554 Évora, Portugal
* Correspondence: ofelia@ipcb.pt

**Abstract:** There is a trend towards the commercialisation of strawberry tree fruit spirit (AUS) with wood ageing, motivated by its favourable sensory characteristics. Additionally, further studies are necessary to elucidate the optimal conditions regarding ageing time and toasting level. This study evaluated the changes in colour and low molecular weight compounds (LMWC) of AUS aged for three and six months using oak wood (*Quercus robur* L.) with light, medium and medium plus toasting levels. For this purpose, phenolic acids (gallic, ellagic, ferulic and syringic acids), phenolic aldehydes (vanillin, syringaldehyde, coniferaldehyde and sinapaldehyde) and furanic aldehydes (furfural, 5-hydroxymethylfurfural and 5-methylfurfural) were quantified using the HPLC method. Chromatic characteristics, colour sensory analysis and total polyphenol index were also analysed. Fourier transform near-infrared spectroscopy (FT-NIR) was used to discriminate between samples. The results emphasized the favourable effect of oak wood contact on enhancing the colour and enriching AUS with low molecular weight compounds (LMWC). AUS aged in medium toasted wood exhibits high levels of total phenolic index, 5-hydroxymethylfurfural, furfural, coniferaldehyde, sinapaldehyde, sum LMWC and chromatic characteristics b* and C. Concentrations of syringaldehyde, ellagic acid, vanillin and syringic acid and a lighter colour (a* chromaticity coordinates) are higher in AUS aged with slightly more toasted wood. Nearly all analysed parameters showed an increase with ageing time. The FT-NIR technique allowed for the differentiation of aged AUS, focusing more on ageing time than on toasting level.

**Keywords:** *Arbutus unedo* spirit; ageing; oak wood; colour; low molecular weight compounds; FT-NIR

## 1. Introduction

The strawberry tree (*Arbutus unedo* L.) is a shrub that originates from the Mediterranean basin but is also found in western Asia, northeastern Africa, the Canary Islands and southern Europe [1,2]. Different parts of the plant (wood, leaves and fruit) have economic importance due to their use in many different commercial applications, namely in the food, pharmaceutical and chemical extractives industries [3–5].

Strawberry tree fruits are usually consumed fresh or in jams or marmalades or used in the production of vinegar and, more traditionally, in alcoholic distillates [6,7]. However,

alcoholic distillate, namely strawberry tree fruit spirit, has gained economic importance due to the improvement of its manufacturing process [8]. Strawberry tree fruits are rich in phenolic compounds (including flavonoids), essential fatty acids, pigments, pectin and vitamins [7]. During the distillation of spirits, the thermal treatment plays a critical role in decomposing these biologically active compounds. This process involves subjecting the fermented fruit to high temperatures, which causes the breakdown of these compounds, diminishing their health benefits [9].

Strawberry tree fruit spirit (AUS) is a traditional beverage consumed in some countries in Europe, mainly Portugal, Spain and Italy; it is produced from the fermentation process (without yeast inoculation) of the whole fruit over several weeks, followed by a distillation usually performed in a copper alembic by a discontinuous process [10]. AUS is colourless, with a clear and bright appearance, and presents fruity and dried fruit odour notes and smooth and soft flavours of strawberry tree fruit [11].

AUS is traditionally consumed without wood ageing; however, more recently, some studies have been developed to understand its quality when wood is used in different periods of ageing [11,12]. Commonly, oak wood is used in the ageing of most kinds of spirits [13,14]. In this context, Galego et al. [12] propose using oak wood for the AUS ageing process, suggesting an ageing time of up to 12 months in wooden barrels subjected to a medium toasting level due to the wood flavour in spirit. Nevertheless, Anjos et al. [11] found that a period of three months is better than six months of alternative ageing using wood fragments with a medium toasting level to keep the characteristic flavour of strawberry tree fruit. This team also obtained the highest score of aroma and overall quality for AUS aged for three months. The application of alternative ageing through the use of wood fragments (staves, chips and cubes, among others) is mainly related to low investments, similar sensory characteristics obtained and lower ageing time [15,16].

The maturation process in wood imparts distinctive sensory attributes to the spirits, raising the consumers' preference [17]. During the ageing period in oak barrels, several physical and chemical transformations occur. They comprise the extraction of compounds from the wood and their reactions with those present in the spirit. Several reactions, like oxidation, esterification, Maillard reactions, polymerization and polycondensation, influence the final sensory profile of the aged spirits [18], contributing in various ways to their organoleptic qualities.

One of the most evident changes observed is the colour of aged spirits, which is influenced by their phenolic compound content [19,20]. Polyphenols are the major compounds extracted from wood that are involved in oxidative reactions and give most of the colour, aroma and taste of wood-aged spirits [21–23]. Phenolic acids, phenolic aldehydes and coumarins are the main polyphenol groups identified in wine spirits ageing in oak wood [19]. In aged wine spirits, the phenolic acids and coumarins contribute to bitter and astringent tastes [24], while phenolic aldehydes are responsible for the typical vanilla and woody notes [25]. During the ageing process, the polymeric phenolic compounds found in oak wood undergo oxidative reactions, forming syringaldehyde and vanillin [26]. However, as far as we know, no studies have been developed on the relationship between the compounds mentioned above and the sensory features of AUS.

On the other hand, furanic aldehydes extracted from wood, such as furfural, hydroxymethyl furfural (HMF) and 5-methyl furfural, contribute to the brown colour and toasted almond aroma of aged spirits [27] and usually increase with the toasting level. The odorant compounds, beverage characteristics and chemical composition of the wood used in the ageing process provided a quality that was different from the spirit drinks [28–30].

The aim of this study was to assess the variation of low molecular weight compounds as well as both analytical and sensory colour during the ageing process of AUS. This was achieved by employing oak wood staves with varying toasting levels over periods of three and six months.

## 2. Materials and Methods

### 2.1. Experimental Design and Samples

The strawberry trees used to produce the AUS grown in the Oleiros municipality in the interior central region of Portugal typically yield a higher quantity of strawberries through organic farming. The strawberry tree, originally native to Portugal, thrives in fields installed in Cambisols without the need for fertilizers or pesticides. Fruits from various orchards in the region are harvested, fermented and then distilled at the Mendes & Mendes, LDA distillery (Oleiros, Portugal). The resulting distillate is blended in large containers to ensure consistency across batches, and a sample from one of these blends was utilized for our research.

The 12 samples of AUS aged with Limousin oak (*Quercus robur* L.) for three and six months were analysed according to the scheme presented in Figure 1.



**Figure 1.** Ageing experimental scheme. LT—light toasting wood; MT—medium toasting wood; MPT—medium plus toasting wood; 3 and 6 represent three and six months with wood contact, respectively.

The assay was performed at a laboratory scale in 2 L glass bottles, and AUS without wood ageing was used as a control with an alcohol strength of 46.2% (*v/v*). Two assay replicates were conducted for each modality under identical conditions, maintaining darkness and controlled temperature (18 °C).

The same AUS was purchased from a certified local producer and used to produce all samples.

The staves were acquired from J. M. Gonçalves cooperage, located in Palaçoulo, Portugal. The different toasting levels were made in the above-mentioned cooperage, using an industrial oven with controlled and differentiated temperature to protect the properties and specific characteristics of oak wood (*Quercus robur* L.). The temperature of the wood surface for the different toast levels was light (115–125 °C), medium (200–210 °C) and medium plus (230 °C to 250 °C).

The quantity of wood staves used in the assay was 85 cm$^2$/L (surface area to volume ratio of a 250 L barrel), to simulate the one usually used in the spirit ageing process. To preserve the sensory characteristics of AUS, a maximum of six months as an ageing time was established, intended to increase the sensory quality of the final product given by wood while maintaining the characteristic aroma and flavour of the AUS.

### 2.2. Production Process of AUS

The production process of AUS involves three steps: harvesting, fermentation and distillation. It begins with carefully harvesting ripe arbutus fruits, typically during late autumn when the fruits are round and red. Only fully matured and healthy fruits are hand-picked to ensure quality.

Once harvested, the fruits undergo washing and sorting to remove any debris or unwanted materials. They are then transferred to 120-litre fermentation tanks without being crushed or inoculated with yeast. Fermentation is initiated naturally by wild yeasts present on the fruit skins. The tanks are sealed with air locks filled with water to allow

for spontaneous alcoholic fermentation with a temperature of around 13–14 °C in a dark environment.

After fermentation, the resulting must undergo distillation in an alembic to increase alcohol strength and produce AUS. The alembic is equipped with a boiler, a 200-litre capacity column, a Swan's neck and a coil submerged in a water condenser. The distilled AUS collected from the column still is then filtered.

The boiler is heated by burning raw wood, and temperatures and pressures during distillation are adjusted based on factors such as the alcohol strength of the mash and the desired characteristics of the final product.

### 2.3. Reagents

Methanol (99.9% *v/v*, LC gradient grade) and formic acid (98% *v/v*, analytical grade) were purchased from Merck (Darmstadt, Germany) and ethanol (99.9% *v/v*, LC gradient grade) was purchased from Carlo Erba (Val de Reuil, France). Ultrapure water (conductivity < 0.055 μS/cm) and distilled water (conductivity < 6.0 μS/cm) were obtained from the Arium Comfort System (Sartorius, Goettingen, Germany). The solutions were prepared fresh prior to use with ethanol/water (75:25, *v/v*). The chemical standards, including their CAS number, purity and supplier, are listed below:

- Gallic acid—CAS Number: 149-91-7; purity of 98%; supplier: Fluka, Buchs, Switzerland;
- 5-Hydroxymethylfurfural—CAS Number: 67-47-0; purity 98%; Fluka, Buchs, Switzerland;
- Furfural—CAS Number: 98-01-1; purity of 98%; supplier: Fluka, Buchs, Switzerland;
- 5-Methylfurfural—CAS Number: 620-02-0; purity of 99%; supplier: Sigma, Steinheim, Germany;
- Syringic acid—CAS Number: 530-57-4; purity of 98%; supplier: Alfa Aesar, Kandel, Germany;
- Ferulic acid—CAS Number: 1135-24-6; purity of 99%; supplier: Fluka, Buchs, Switzerland;
- Ellagic acid—CAS Number: 476-66-4; purity of 98%; supplier: Fluka, Buchs, Switzerland;
- Vanillin—CAS Number: 121-33-5; purity of 99%; supplier: Fluka, Buchs, Switzerland;
- Syringaldehyde—CAS Number: 134-96-3; purity of 98%; supplier: Sigma, Steinheim, Germany;
- Coniferaldehyde—CAS Number: 458-36-6; purity of 98%; supplier: Sigma, Steinheim, Germany;
- Sinapaldehyde—CAS Number: 4206-58-0; purity of 98%; supplier: Sigma, Steinheim, Germany.

### 2.4. Chromatic Characteristics

Transmittance measurement was made every 10 nm from 380 to 770 nm, using a D65 illuminant and a 10° standard observer.

A Varian Cary 100 Bio spectrophotometer (Santa Clara, CA, USA) was used to measure the CIELab chromatic characteristics of AUS with a 10-mm glass cell according to the methodology used by Canas et al. [19]. The colour parameters measured included lightness (L*), a* and b* chromaticity coordinates, chroma and browning, determined by the absorbance at 470 nm. Duplicate analyses were conducted for all samples.

### 2.5. Colour Sensory Analysis

The colour of AUS samples with 3 and 6 months of ageing was assessed by quantitative descriptive analysis employing a panel of ten tasters carefully chosen, trained and experienced with this kind of spirit [11]. This tasting panel consisted of eight judges, evenly split between genders and ranging in age from 27 to 56 years.

The sensory sessions were conducted in an INIAV sensory room in natural light and with a room temperature of 20 °C. Each taster received the whole set of samples, served in standardized wine-tasting glasses (ISO standard 3591) [31], in a carefully balanced sequence to minimize potential carryover effects and anonymized with randomly assigned three-digit codes. Tasters were instructed to assess the intensity of the colour attributes

(yellow-green, yellow-straw, golden, amber and greenish) with a structured scale ranging from 0 to 5, indicating no perception to strong perception, respectively [32].

### 2.6. Total Polyphenol Index

The total polyphenol index (TPI) was determined following the methodology outlined by Cetó et al. [33] with some modifications. Briefly, the samples (AUS) were first diluted with a mixture of ethanol and water in a 47:53 $v/v$ ratio. Subsequently, the absorbance was measured using a Varian Cary 100 Bio spectrophotometer (Santa Clara, CA, USA) equipped with a 10 mm quartz cell at a wavelength of 280 nm. TPI was then calculated by multiplying the absorbance obtained by the dilution factor.

### 2.7. Low Molecular Weight Compounds Analysis

The low molecular weight compound content was quantified in AUS samples by the HPLC method. The HPLC Lachrom Merck Hitachi system (Merck, Darmstadt, Germany) contains a quaternary pump L-7100; a column oven L-7350; and a Hitachi UV-Vis detector L-7400 and autosampler L-7250. The chromatographic separation was performed on a LiChrospher RP 18 (5 μm) (5 μm, 250 mm × 4 mm ID) column (Merck, Darmstadt, Germany).

The mobile phase was comprised of water: formic acid (98:2, $v/v$) as eluent A and methanol: water: formic acid (70:28:2, $v/v/v$) as eluent B at a flow rate of 1 mL/min and column temperature of 40 °C. The solvents (aqueous and methanol) used in chromatography conditions were filtered through 0.45 μm membranes (HATF and FH types, respectively; Millipore, Bedford, MA, EUA) prior to analysis. The following gradient program was used: 0–3 min at 0% isocratic B; 3–25 min from 0% to 40% B; 25–43 min from 40% to 60% B; 43–55 min at 60% isocratic B; 55–60 min from 60 to 80% A; 60–65 min at 80% isocratic B; 65–75 min from 80 to 0% B and finally returning to the initial conditions.

The detection of furanic aldehydes and phenolic acids was performed at 280 nm and of phenolic aldehydes at 320 nm. Phenolic acids (ellagic, ferulic, gallic and syringic acids), phenolic aldehydes (vanillin, syringaldehyde, coniferaldehyde and sinapaldehyde) and furanic aldehydes (5-hydroxymethylfurfural, furfural and 5-methylfurfural) were analysed and quantified through calibration curves (mg/L), according to a validated methodology [34]. The HPLC coupled software used was HSM D-7000 (version 3.1) for data management, acquisition and treatment (Merck, Darmstadt, Germany). All samples were directly analysed by injecting 20 μL of beverage with 20 mg/L of 4-hydroxybenzaldehyde (internal standard) and filtered through a 0.45 μm membrane (Filter-Bio, Nantong, China). The compounds were identified by comparing their retention times and peak shapes to those of known compounds (standards) using the same chromatographic system. Quantification of compounds was carried out using calibration curves established with the respective commercial standards. All analyses were made in duplicate, and the results were expressed as mg/L absolute ethanol (mg/L AE).

### 2.8. Vibrational Spectroscopy

The AUS samples' spectra were acquired utilizing an FT-NIR spectrometer (MPA Bruker) operating in transmitted light mode. Measurements were conducted at room temperature with 1 mm quartz cells. The spectral resolution was set at 8 cm$^{-1}$, with 32 scans within the wavenumber range from 12,500 to 4000 cm$^{-1}$ [35,36]. After scanning a sequence of 10 samples, a background with air was achieved.

### 2.9. Data Analysis

A two-way analysis of variance (ANOVA) was conducted with two fixed factors: (1) toasting level-TL (with three levels: light toasting wood, medium toasting wood and medium plus toasting wood); (2) ageing time—T (with two levels: 3 and 6 months). Scheffe's test was applied for mean comparison of the low molecular weight compound contents in the aged AUS when a significant difference ($p < 0.05$) was detected.

ANOVA was performed, excluding the control samples due to their higher differences. The distinct composition of these control samples caused clustering with both the control and aged samples. Consequently, excluding the control sample from the ANOVA is essential for accurately assessing the impact of ageing time and wood toasting levels. The control sample was used only to understand the compounds already present in the beverage or those originating from wood contact.

The ANOVA and heat maps were analysed using Statistica version 7.0 software (Statsoft Inc., Tulsa, OK, USA). The spectral data acquired with FT-NIR underwent a PCA analysis using UnscramblerX 10.5 (CAMO, Oslo, Norway) to distinguish between samples. Five replicated spectra were collected for each sample.

## 3. Results and Discussion

The mean values of the chromatic characteristics and total polyphenol index (TPI) of AUS over time are presented in Table 1. The results for chromatic characteristics of the aged AUS reveal an evolution of lightness (L*), green-red hue (a*), yellow hue (b*), chroma (C) and brown hue (A470) with the ageing time. Concerning the toasting level, a more pronounced colour evolution was observed in aged spirits using a medium toasting level for six months. This behaviour was expected and consistent with previous studies on aged wine spirits, in which the medium toasting level induced higher TPI values [19].

**Table 1.** Effect of the toasting level on the chromatic characteristics and total polyphenol index of the AUS after 3 and 6 months of ageing.

| Samples | L | a* | b* | C | A470 | TPI |
|---|---|---|---|---|---|---|
| C | $100.00 \pm 0.06$ | $-0.08 \pm 0.01$ | $0.55 \pm 0.01$ | $0.56 \pm 0.00$ | $100.02 \pm 0.06$ | -- |
| LT3 | $96.57 \pm 0.1$ [ab] | $-2.44 \pm 0.02$ [a] | $17.29 \pm 0.52$ [a] | $17.46 \pm 0.52$ [a] | $88.23 \pm 0.39$ [b] | $7.17 \pm 0.06$ [a] |
| MT3 | $95.20 \pm 3.86$ [a] | $-2.14 \pm 0.39$ [ab] | $36.35 \pm 1.83$ [bc] | $36.42 \pm 1.81$ [bc] | $77.47 \pm 5.64$ [a] | $18.51 \pm 0.41$ [bc] |
| MPT3 | $97.14 \pm 0.66$ [ab] | $-1.47 \pm 0.29$ [b] | $39.48 \pm 2.29$ [bc] | $39.51 \pm 2.28$ [bc] | $78.13 \pm 2.01$ [a] | $17.72 \pm 1.09$ [b] |
| LT6 | $99.94 \pm 2.17$ [b] | $-2.61 \pm 0.40$ [a] | $31.96 \pm 9.97$ [b] | $32.24 \pm 9.75$ [b] | $85.53 \pm 3.65$ [b] | $15.20 \pm 5.16$ [b] |
| MT6 | $97.19 \pm 1.55$ [ab] | $-1.67 \pm 0.90$ [b] | $41.99 \pm 4.84$ [c] | $42.04 \pm 4.79$ [c] | $76.94 \pm 4.63$ [a] | $23.98 \pm 2.77$ [c] |
| MPT6 | $97.87 \pm 1.13$ [ab] | $-1.54 \pm 0.46$ [b] | $37.88 \pm 3.72$ [bc] | $37.92 \pm 3.70$ [bc] | $79.84 \pm 3.31$ [a] | $18.41 \pm 1.71$ [bc] |
| TL | n.s. | 41.7 ** | 46.4 ** | 45.8 *** | 44.9 ** | 57.1 *** |
| T | 23.5 * | n.s. | 12.1 * | 12.5 * | n.s. | 23.6 *** |
| TLxT | n.s. | n.s. | 18.2 * | 18.9 * | n.s. | n.s. |
| R | 76.5 | 58.3 | 23.3 | 22.8 | 55.1 | 19.3 |

For each compound, means within the same column followed by different letters are significantly different ($p < 0.05$); n.s. for $p > 0.05$; * $0.01 < p < 0.05$; ** $0.001 < p < 0.01$; *** $p < 0.001$; L—lightness; a* and b*—CIELab coordinates; C—chroma; A470—absorbance at 470 nm; n.s.—without significant effect.

Regarding the chromatic characteristics, lightness, which reflects the AUS transparency [37], was lower in the spirit aged 3 months than for 6 months (97.19% to 99.94%). Furthermore, MT3 was significantly lower than LT6, and other samples showed no difference. Only time is a significant factor explained by the total variation, and the variability observed is higher and explains 76.50% of the total variance. This variability could be explained by the more pronounced wood variability when a laboratory trial was used rather than an industrial one.

a* takes negative values for the green hue and positive values for the red one. The aged spirits had a slight green hue ($-2.44$ to $-1.47$), and this tonality decreased with the increase in the toasting level. For coordinate a*, only the toasting level was significant and explained 58.3% of the total variance; the remaining variance is explained by the variability between samples. The variation in green colour could be explained by the presence of coumarins (umbelliferone and scopoletin) from oak wood, as stated by Canas et al. [38] for aged wine

spirits. Umbelliferone and scopoletin have been quantified in oak *(Quercus robur* L.) wood. Coumarins have UV-Vis absorption up to 420 nm, which can originate green attributes in the early stages of ageing because they do not absorb between 500 and 565 nm [39,40]. The decrease in green colouration observed in samples with the increase of toasting level of wood is due to the thermal sensitivity of coumarins [41].

As expected for aged spirits, the yellowness (b* values) increased as the ageing time and toasting level increased. Indeed, the toasting level and ageing time were significant and explained 46.4% and 12.1% of the total variance, respectively. Consequently, a similar trend was observed in C values, where both the toasting level and ageing time were found to be significant, explaining 45.8% and 12.5% of the total variance, respectively. The b* and C values were higher for MT6. These results were reported in previous works regardless of the ageing technology [19], which suggests that coordinate b* has a more pronounced impact on chroma than coordinate a*. It is important to note that TPI values were higher for MT3 and MT6 and lower for LT3 and LT6, confirming that the low molecular weight compounds extracted from the wood contribute to increased b* values.

Regarding the MT and MPT samples, the golden attribute remains quite similar after 3 and 6 months of ageing (mean score of 2.9 to 3.2). However, the AUS samples aged with LT wood exhibit a more pronounced yellow-straw hue after 6 months (mean score of 0.2) and a less intense golden attribute compared to the other toasting levels (mean score of 1.1 for 3 months of ageing and 1.6 for 6 months of ageing) (Figure 2). The amber attribute shows a greater intensity in samples aged with MPT toasting levels for both 3 and 6 months of ageing, with mean scores of 1.2 and 1.4, respectively.

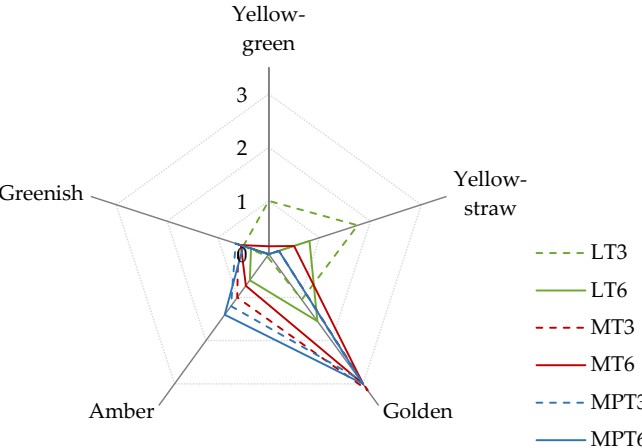

**Figure 2.** Colour profile based on mean scores of panel attributes for the AUS.

Differentiation made by the tasters based on the colour attributes of AUS samples mainly resulted from the evolution imparted by the toasting level rather than the ageing time. Furthermore, it is important that the ageing period in this kind of beverage is lower to ensure the characteristic flavour of the original spirit *(Arbutus unedo* Spirit) [11]. As previously reported [19], the different AUS samples observed a consistency between the sensory attributes evaluated by the expert panel and the chromatic characteristics.

Throughout the ageing process, the AUS gradually became enriched with wood-extracted compounds, particularly those of lower molecular weight. This led to an increase in TPI over time, with a more pronounced effect observed associated with the medium toasting level (from 18.51 to 23.98). According to the ANOVA results, the various toasting levels explain 57% of the total observed variance, while the ageing time explains 23.6% of the total variance (Table 1).

Table 2 exhibits the variation of low molecular weight compounds (LMWC) observed in the studied AUS samples, while Table 3 presents the ANOVA results considering the effects of ageing time and toasting level. The variation observed is mainly due to the differences in the toasting level (Table 2), with a percentage of variance ranging from 50.5%

for gallic acid to 99.3% for ellagic acid (Table 3). The ageing time significantly influences the levels of 5-hydroxymethylfurfural, syringic acid, ferulic acid, vanillin, syringaldehyde, coniferaldehyde, sinapaldehyde and the sum of low molecular weight compounds (sum-LMWC), accounting for 13.6%, 3.2%, 29.1%, 16.1%, 12.9%, 1%, 3.7% and 8.7% of the total variance, respectively. (Tables 2 and 3).

**Table 2.** Effect of the toasting level on the contents of low molecular weight compounds (mg/L AS) of the AUS after 3 and 6 months of ageing.

| Samples | C | LT3 | MT3 | MPT3 | LT6 | MT6 | MPT6 |
|---|---|---|---|---|---|---|---|
| AS (% *v/v*) [11] | 46.2 ± 0.00 [a] | 47.9 ± 0.11 [c] | 46.9 ± 0.01 [b] | 46.9 ± 0.01 [b] | 46.8 ± 0.13 [b] | 47.0 ± 0.03 [b] | 47.0 ± 0.03 [bc] |
| **HMF** | nd | 2.41 ± 0.85 [a] | 14.54 ± 0.36 [b] | 4.86 ± 1.26 [a] | 7.44 ± 0.44 [a] | 19.84 ± 6.86 [b] | 7.65 ± 0.29 [a] |
| **Furf** | 6.07 ± 0.06 [a] | 12.19 ± 1.52 [b] | 41.44 ± 1.43 [c] | 47.66 ± 7.95 [c] | 17.97 ± 1.91 [b] | 56.73 ± 22.52 [c] | 37.20 ± 1.55 [c] |
| **5Mfurf** | nd | 0.43 ± 0.07 [a] | 2.43 ± 0.29 [c] | 1.66 ± 0.02 [b] | 0.61 ± 0.04 [a] | 3.96 ± 1.84 [c] | 1.79 ± 0.08 [b] |
| **Gall** | nd | 1.67 ± 0.42 [a] | 7.60 ± 3.15 [b] | 0.58 ± 0.48 [a] | 9.86 ± 6.56 [b] | 7.79 ± 2.07 [b] | 0.02 ± 0.09 [a] |
| **Syrg** | nd | 1.16 ± 0.01 [a] | 2.41 ± 0.34 [b] | 9.83 ± 0.22 [c] | 1.38 ± 0.05 [a] | 3.80 ± 0.60 [b] | 12.41 ± 1.15 [d] |
| **Ferul** | nd | 0.07 ± 0.01 [a] | 0.05 ± 0.00 [a] | 0.08 ± 0.03 [a] | 0.16 ± 0.06 [b] | 0.05 ± 0.00 [a] | 0.14 ± 0.01 [b] |
| **Ellag** | nd | 2.53 ± 0.09 [a] | 4.40 ± 0.34 [ab] | 8.30 ± 0.17 [c] | 4.56 ± 0.40 [ab] | 5.32 ± 0.01 [b] | 7.16 ± 1.57 [c] |
| **Vanil** | nd | 0.88 ± 0.04 [a] | 3.94 ± 0.26 [c] | 3.45 ± 0.21 [c] | 1.51 ± 0.07 [b] | 4.77 ± 0.15 [d] | 5.71 ± 0.19 [e] |
| **Syrde** | nd | 0.90 ± 0.08 [a] | 5.17 ± 1.68 [b] | 11.42 ± 0.78 [d] | 1.86 ± 0.29 [a] | 8.80 ± 0.05 [c] | 18.88 ± 0.24 [e] |
| **Cofde** | nd | 2.26 ± 0.25 [ab] | 12.64 ± 0.81 [c] | 1.52 ± 0.13 [a] | 3.09 ± 0.51 [b] | 15.06 ± 1.21 [d] | 1.42 ± 0.26 [a] |
| **Spide** | nd | 2.36 ± 0.36 [a] | 22.00 ± 3.68 [b] | 4.34 ± 0.00 [a] | 4.64 ± 0.23 [a] | 30.89 ± 0.18 [c] | 4.59 ± 0.77 [a] |
| **Sum-LMWC** | nd | 26.89 ± 13.67 [a] | 116.66 ± 58.43 [bc] | 93.75 ± 47.07 [b] | 53.10 ± 27.22 [a] | 157.06 ± 81.88 [c] | 97.02 ± 48.52 [b] |

For each compound, means within the same row followed by different letters are significantly different (*p* < 0.05); nd—not detected; AS—alcohol strength; Gall—gallic acid; HMF—5-hydroxymethylfurfural; Furf—furfural; 5Mfurf –5-methylfurfural; Syrg—syringic acid; Ferul—ferulic acid; Ellag—ellagic acid; Vanil—vanillin; Syrde—syringaldehyde; Cofde—coniferaldehyde; Spide—sinapaldehyde; sum-LMWC—sum of the low molecular weight compounds; nd—not detected.

**Table 3.** Percentage of variance based on ANOVA results for the contents of low molecular weight compounds (mg/L AE) of the AUS after 3 and 6 months of ageing.

| Samples | TL (%) | T (%) | TLxT (%) | R (%) |
|---|---|---|---|---|
| **HMF** | 69.0 *** | 13.6 ** | n.s. | 17.4 |
| **Furf** | 70.5 *** | n.s. | n.s. | 29.5 |
| **5Mfurf** | 68.6 *** | n.s. | n.s. | 31.4 |
| **Gall** | 50.5 ** | n.s. | n.s. | 49.5 |
| **Syrg** | 93.4 *** | 3.2 *** | 2.0 ** | 1.4 |
| **Ferul** | 24.2 ** | 29.1 ** | 20.8 * | 25.9 |
| **Ellag** | 99.3 *** | n.s. | n.s. | 0.7 |
| **Vanil** | 75.0 *** | 16.1 *** | 8.1 *** | 0.8 |
| **Syrde** | 77.4 *** | 12.9 *** | 8.3 *** | 1.3 |
| **Cofde** | 96.4 *** | 1.0 ** | 1.4 * | 1.2 |
| **Spide** | 89.6 *** | 3.7 *** | 5.0 *** | 1.7 |
| **Sum-LMWC** | 81.7 *** | 8.7 ** | n.s. | 9.6 |

n.s. for *p* > 0.05; * 0.01 < *p* < 0.05; ** 0.001 < *p* < 0.01; *** *p* < 0.001; Gall—gallic acid; HMF—5-hydroxymethylfurfural; Furf—furfural; 5Mfurf—5-methylfurfural; Syrg—syringic acid; Ferul—ferulic acid; Ellag—ellagic acid; Vanil—vanillin; Syrde—syringaldehyde; Cofde—coniferaldehyde; Spide—sinapaldehyde; sum-LMWC—sum of the low molecular weight compounds; TL—toasting level; T—ageing time; R—residue.

Furanic aldehydes were primarily influenced by the toasting intensity (TL; 68.6 to 70.5% of the total variation, Table 3), whose concentration increased from light to medium toasted wood (Table 2). However, the AUS samples exhibited a decrease in furanic compounds from medium to medium-plus toasting levels, except for furfural amounts after 3 months, which remained unchanged (Table 2). These results are in accordance with those obtained for HMF in aged wine spirits [42]. The ageing time only influenced the HMF contents, which were higher in the samples aged for 6 months (7.44, 19.4 and 7.65 mg/L AE for LT6, MT6 and MPT6, respectively). That variation accounts for 13.6% of the total variance (Tables 2 and 3). Furfural content has been used to determine and control the toasting degree; therefore, heating of oak wood increases the amounts of furanic compounds up to a certain toasting level, but higher toasting tends to decrease these compounds due to the pyrolytic effect [43]. Regarding phenolic acids, the amounts of ellagic and gallic acid were only significantly influenced by the toasting level, with 50.5% and 99.3% of the total variation, respectively (Table 3). The levels of gallic acid decrease with an increase in the toasting level for 6 months of ageing (9.86 < 7.79 < 0.02 mg/L AE), with the lowest levels observed in the AUS aged with medium plus toasted wood after 6 months of ageing (0.02 mg/L AE) (Table 2). For ellagic acid, the opposite effect is observed, with its content increasing with the rise of toasting level (2.53 < 4.40 < 8.30 mg/L AE for 3 months and 4.56 < 5.32 < 7.16 mg/L AE for 6 months). Similar results were obtained by Canas et al. [42] in aged wine spirits with chestnut wood. Gallic acid is extremely sensitive to heat treatment, systematically decreasing its content in the wood when exposed to high temperatures during toasting (MPT). Conversely, ellagic acid increases in relation to toasting intensity resulting from the thermo-degradation of ellagitannins [44].

For the other analysed phenolic acids, such as syringic (3.3% of the total variance) and ferulic (29.1% of the total variance), a significant effect of toasting level and ageing time was observed. Moreover, these factors had a significant interaction (2.0% and 20.8% of the total variance, respectively) (Table 3). Therefore, syringic acid tends to increase with the increase of toasting level (1.16 to 9.83 mg/L AE to 3 months and 1.39 to 12.41 mg/L AE to 6 months), and this increase was more pronounced after 6 months of ageing (Table 2). For ferulic acid, the lowest levels (0.05 mg/L AE) were found in the samples aged with medium toasting wood (MT), while the highest levels were found in those aged with LT and MPT; this differentiation was more evident after 6 months. Although certain amounts of phenolic acids could also be influenced by fermentation procedures, as noted by Lisov et al. in wines [45], the results shown in Table 2 indicate that these acids in AUS originate solely from wood contact, as they are not detected in the unaged sample (C). Regarding phenolic aldehydes, a significant effect of the toasting level is observed, together with an effect of the ageing time (from 75% to 94.4% of the total variance) and an interaction of both factors (from 1.4% to 8.3% of the total variance) (Table 3). Vanillin and syringaldehyde levels increase with the rise of toasting level, especially in samples aged with wood for 6 months, where an increase from 1.51 to 5.71 mg/L AE was observed for vanillin and an increase from 1.86 to 18.88 mg/L AE was observed for syringaldehyde (Table 2). Other authors achieved similar results when working with different types of spirits [46–48]. However, coniferaldehyde and sinapaldehyde increase from light to medium toasting, followed by a sharp decrease for medium plus toasting levels. This behaviour was more evident after 6 months of ageing. The lowest content of coniferaldehyde was observed for MPT6 (1.42 mg/L AE), while the highest was observed for MT6 (15.06 mg/L AE). According to Le Floch et al. [49], throughout the toasting process, lignin undergoes transformation, yielding two distinguishable sets of phenolic aldehydes. One group comprises guaiacyl-type compounds, including coniferaldehyde and vanillin, while the other group comprises syringyl-type compounds, such as sinapaldehyde and syringaldehyde. The oxidation of sinapaldehyde gives rise to syringaldehyde, which is oxidized to syringic acid; similarly, coniferaldehyde is converted to vanillin [46]. These reactions of oxidation can explain the reduction of sinapaldehyde and coniferaldehyde in the samples aged with medium plus

toasting wood (MPT), configuring the increase of the syringaldehyde, syringic acid and vanillin (Table 2).

For the comprehensive physicochemical assessment, a holistic analysis was performed using a cluster heat map, qualifying the correlation of characteristics with the ageing technologies under investigation (Figure 3).

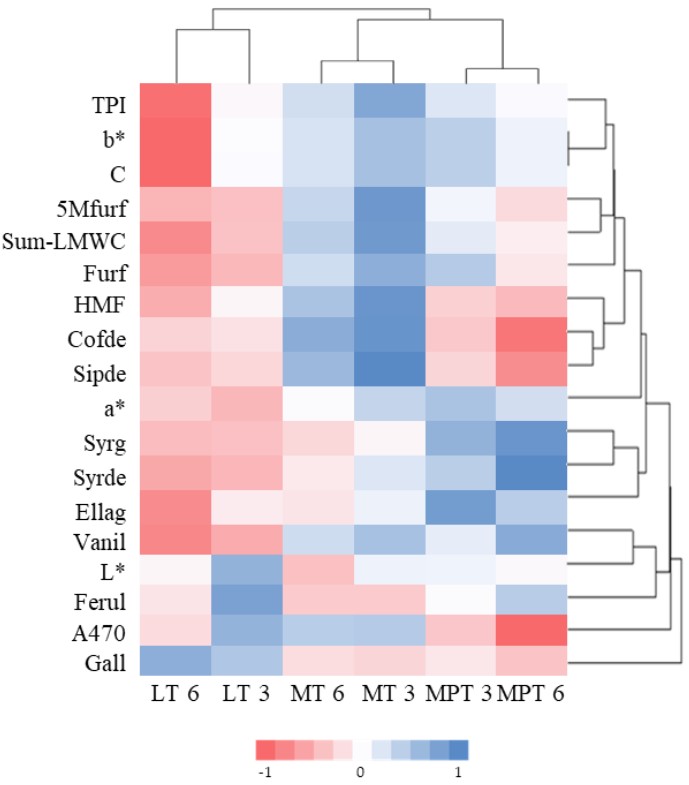

**Figure 3.** Heat map plotting clusters of low molecular weight compounds according to the AUS ageing with different toasting levels of oak wood during 3 and 6 months. TPI—total polyphenol index; Gall—gallic acid; HMF—5-hydroxymethylfurfural; Furf—furfural; 5Mfurf—5-methylfurfural; Syrg—syringic acid; Ferul—ferulic acid; Ellag—ellagic acid; Vanil—vanillin; Syrde—syringaldehyde; Cofde—coniferaldehyde; Spide—sinapaldehyde; sum-LMWC—sum of the low molecular weight compounds; L*—lightness; C—chroma; a* and b*—chromaticity coordinates; A 470—absorbance at 470 nm. LT—light toasting wood; MT—medium toasting wood; MPT—medium plus toasting wood; 3 and 6 represent three and six months with wood contact, respectively.

The heat map clearly split the AUS into three groups based on the toasting level of the wood used (light toasting, medium toasting and medium-plus toasting wood). These findings align with and strengthen the ANOVA results outlined in Table 3.

The key analytical parameters for distinguishing the AUSs according to the toasting level used were: total phenolic index; 5-hydroxymethylfurfural; furfural; 5-methylfurfural; coniferaldehyde; sinapaldehyde; the sum of the low molecular weight compounds; and the colour parameters b* and C. The beverage aged in medium toasting wood is the one that presents higher positive values for the aforementioned analytical parameters.

Higher values of syringaldehyde, ellagic acid, vanillin, syringic acid and lightness a* chromaticity coordinates are distinctive for the samples aged with medium plus toasting wood.

Figure 4 shows an AUS NIR spectrum, which is similar to those obtained by other authors for fruit spirits and wine spirits [35,50,51].

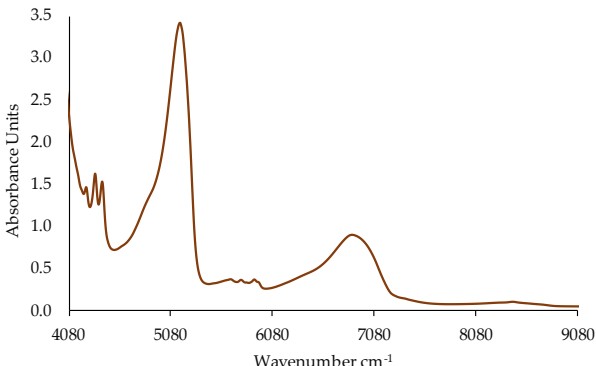

**Figure 4.** Normalized absorption spectra of representative AUS samples acquired with FT-NIR.

The spectral bands in the AUS spectra closely resemble those identified by Anjos et al. [35] in wine spirits, although with slight differences. The most significant bands identified in this matrix are presented below [36,52–55]:

- $5176 \ cm^{-1}$: combination of O–H group stretching and deformation, alongside first overtones of water and ethanol, as well as C–H stretch first overtones;
- $4412 \ cm^{-1}$: ethanol, sugars and phenolic compound absorption band;
- $4337 \ cm^{-1}$: second overtone of stretching C–H and O–H vibrations, often associated with functional groups found in alcohols and carboxylic acids. The two previous bands are characteristics of samples with methanol;
- $4258 \ cm^{-1}$: the combination of stretching and bending deformation of C–H units;
- $5689 \ cm^{-1}$ (three small peaks): ascribed to the C–H stretch of the first overtones of $CH_2$ and $CH_3$ groups and O–H from aromatic groups;
- $6896$–$8434 \ cm^{-1}$ (peak with lower intensity): second overtone of the C–H stretch of ethanol and the combination of the bending vibration of O–H bend and the first overtone of the stretching O–H related to the water influence.

The infrared spectral region from $12,000 \ cm^{-1}$ to $4000 \ cm^{-1}$ was acquired; however, only $4080 \ cm^{-1}$ and $8080 \ cm^{-1}$ have important information for PCA analyses.

Figure 5A shows the score plot of two principal components made with FT-NIR spectra for AUS samples, taking into account the wood toasting levels and the ageing time. The first two principal components accounted for 97% of the total variance, in which the first component (92% of the total variance) divided the samples according to the ageing time.

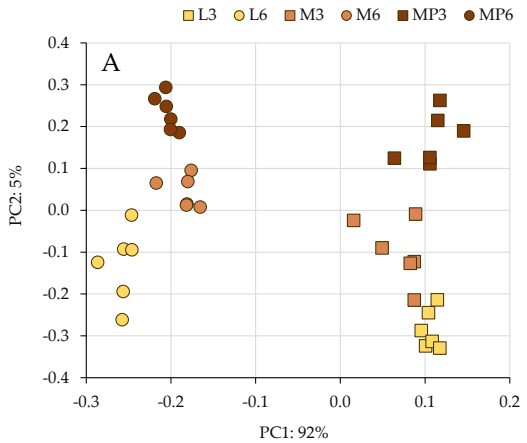
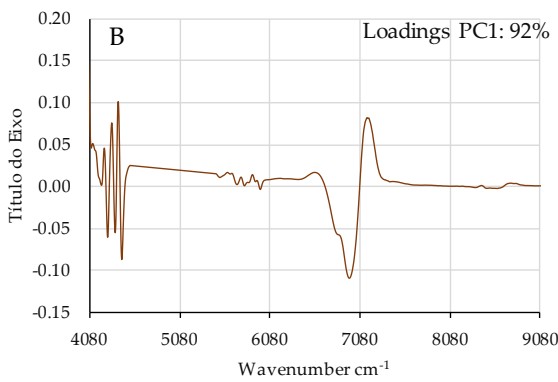

**Figure 5.** Principal components analysis of AUS results performed with the spectral data acquired with FT-NIR, concerning the toasting level of the wood used in the ageing process and ageing period. (**A**)—Score plot, (**B**)—loading plot. LT—light toasting wood; MT—medium toasting wood; MPT—medium plus toasting wood; 3 and 6 represent three and six months with wood contact, respectively.

The ANOVA results shown in Tables 1 and 2 highlighted the impact of the toasting level on both the colour and chemical characteristics of the AUS samples. Additionally, an upward trend was observed in the concentration of certain low molecular weight compounds over the ageing period.

The differentiation between toasting levels is also evident, but only through PC2, which explains only 5% of the total variation.

The prominent regions explaining the observed variation in AUS samples were between 4080 cm$^{-1}$ to 8080 cm$^{-1}$. Despite the significant influence of toasting level, the impact of ageing time was also noticeable, supporting the observed differences and the potential utility of FT-NIR to characterise and discriminate such samples.

As reported by Anjos et al. [11] for FTIR-ATR, our findings using FT-NIR further underline the robustness of these techniques in distinguishing AUS aged with wood based on toasting level and ageing time. The validation of the FT-NIR technique for developing calibration models of certain compounds present in alcoholic beverages was also confirmed by other authors [35,50].

### 4. Conclusions

The trial conducted with *Arbutus unedo* spirit (AUS) aged with wood fragments subjected to three toasting levels for a maximum period of 6 months revealed that toasting level and ageing time are significant, influential factors in the composition of the obtained AUS, with the former being the more discriminative factor.

The results confirmed the positive influence of oak wood contact on the colour and enrichment of AUS in low molecular weight compounds, especially by using fragments with medium toasting or medium plus toasting.

Moreover, the FT-NIR technique enabled the discrimination of aged AUS based on ageing time and toasting level and the most promising regions for sample differentiation were identified.

The results obtained indicate that AUS producers may use different ageing procedures in the future, to design new products.

**Author Contributions:** Conceptualization. O.A. and I.C.; methodology. O.A. and I.C.; formal analysis. I.C., S.C., C.A.L.A. and. O.A.; investigation. O.A., I.C., S.O.-A., C.A.L.A. and S.C; resources. O.A., S.C. and I.C.; writing—original draft preparation. O.A. and C.A.L.A.; review. O.A., I.C., S.O.-A., C.A.L.A. and S.C; editing. O.A., S.C. and I.C.; project administration. O.A. and I.C.; funding acquisition. O.A., S.C. and I.C. All authors have read and agreed to the published version of the manuscript.

**Funding:** This work was supported by Foundation for Science and Technology (FCT, Portugal) for their financial support through national funds FCT/MCTES (PIDDAC) to CERNAS-IPCB, UIDB/00681/2020 (DOI: 10.54499/UIDP/00681/2020); CEF, UIDB/00239/2020 (DOI: 10.54499/UIDB/00239/2020) and TERRA (DOI: 10.54499/LA/P/0092/2020); MED, UIDB/05183/2020.

**Institutional Review Board Statement:** Not applicable.

**Informed Consent Statement:** Not applicable.

**Data Availability Statement:** Data are contained within the article.

**Acknowledgments:** Authors would like to extend their gratitude to the dedicated tasters of the INIAV sensory panel for their consistent participation and accessibility. The authors thank Mendes & Mendes, LDA distillery for the *Arbutus unedo* distillate supplied.

**Conflicts of Interest:** The authors declare no conflicts of interest.

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
