# Peer review of "Characterisation of Low Molecular Weight Compounds of Strawberry Tree (Arbutus unedo L.) Fruit Spirit Aged with Oak Wood"

_fermentation, doi:10.3390/fermentation10050253_

Round 1

Reviewer 1 Report

Comments and Suggestions for Authors

I think the article needs to be improved with necessary and missing informations in materials and method, so the obtained results can be discussed. If authors can provide the missing informations then the article can be reviewed.

Introduction

Line 48: correct Europa with Europe

Line 62: correct qualities with quality

Line 78: I think the word "impart" is not necessary in the sentence 

Materials and Methods

Paragraph 2.1: you should reporte the roasting temperature. Light, medium and medium plus without specific temperatures are not scientific data

Line 105: controlled temperature I think is better than controlling

Lines 119-120: you should re-write the sentence "Transmittance measurement was made every 10 nm from 380 to 770 nm, using a D65 119 illuminant and a 10° standard observer., were assessed according to." According to what?

In Paragraph 2.5 a lot of informations are missed, you must report: the precise column characteristic (lenght), the mobile phases, the gradient conditions, the column flow, what material was the filter? How did you identified all the compounds?

Table 1: why did not you considere the C sample in the statistical analysis?

Comments on the Quality of English Language

Extensive editing of English language required

Author Response

The author sincerely appreciates your thoughtful revision of the manuscript, which has enhanced the paper's quality and interest. All the changes and suggestions have been duly considered. The responses can be found in the attached document.

Reviewer 2 Report

Comments and Suggestions for Authors

Dear Authors,

This study is dealing with the investigation of chemical profile of strawberry tree fruit spirit which was aged during different periods with oak woods. Some finding is possible to apply in practice. In this manuscript are missing important information.

Manuscript missing  the most important results. Insert values of results in manuscript.

Highlight in introduction that thermal treatment applied during the distillation of spirit is responsible for decomposition of biologically active compounds which are present in strawberry tree fruit.

What was the alcohol content of strawberry tree fruit spirit before ageing? Insert it in manuscript. It is very important.

What was the alcohol content of strawberry tree fruit spirit after 3 and 6 month of ageing during three different toasting procedures? Insert it in manuscript. It is very important.

In the section 2 insert new subsection in which you will highlight following information regarding strawberry tree fruit:

What is the origin of strawberry tree fruit city, region insert coordinates?

Did you use any fertilizers during the growing of this fruit?

Did you use any substances for protection against pests?

On what kind of soil fruit was grown?

In the section 2 insert new subsection in which you will briefly highlight production process of spirit obtained from this fruit. What are physic-chemical characteristics of that spirit?

What kind of detector was used in HPLC system? Insert it in manuscript.

In the subsection 2.5. are missing conditions (ionization mode ESI, cone voltage, collision energy) which you applied during the quantification of selected compounds. Insert it. It is important.

In the section 2 is missing subsection Chemicals and reagents. Insert it.

Highlight in the discussion that different technological approach applied during the winemaking procedure significantly affects on the content of phenolic compounds such as gallic and syringic acids. Kindly consider to cite Fermentation 9(7), (2023), 695.

Wish you all the best in the future work,

Author Response

(The authors gave the same response as above.)

Round 2

Reviewer 1 Report

Comments and Suggestions for Authors

I thank the authors for the implementation of the article.

Just one minor revision:

Paragraph 2.7: identification of tocols is still missed

Author Response

Dear reviewer,

Thanks very much for your carefully revise of the manuscript. 

Please see the responses in the attached document

Reviewer 2 Report

Comments and Suggestions for Authors

Dear Authors, 

Thank you very much for revised version of manuscript an all answers on my questions. i have made one typing mistake in review which I have sent you before. I meant that abstract is missing the most important results, so insert results in abstract. 

Wish you all the best i nfuture work, 

Author Response

(The authors gave the same response as above.)
